# OxDNA to Study Species Interactions

**DOI:** 10.3390/e24040458

**Published:** 2022-03-26

**Authors:** Francesco Mambretti, Nicolò Pedrani, Luca Casiraghi, Elvezia Maria Paraboschi, Tommaso Bellini, Samir Suweis

**Affiliations:** 1Dipartimento di Fisica e Astronomia, Università degli Studi di Padova, Via Marzolo 8, 35131 Padova, Italy; n.pedrani@campus.unimib.it (N.P.); samir.suweis@unipd.it (S.S.); 2Atomistic Simulations, Italian Institute of Technology, Via Melen 83, 16152 Genova, Italy; 3Department of Materials Science, Università di Milano-Bicocca, Via Cozzi 55, 20125 Milano, Italy; 4Dipartimento di Biotecnologie Mediche e Medicina Traslazionale, Università degli Studi di Milano, Via Fratelli Cervi, 93-L.I.T.A., 20054 Segrate, Italy; luca.casiraghi@unimi.it (L.C.); tommaso.bellini@unimi.it (T.B.); 5Department of Biomedical Sciences, Humanitas University, Via Levi Montalcini 4, 20072 Pieve Emanuele, Italy; elvezia_maria.paraboschi@hunimed.eu; 6Humanitas Clinical and Research Center, IRCCS, Via Manzoni 56, 20089 Rozzano, Italy

**Keywords:** ecological competition, single-stranded DNA, molecular dynamics

## Abstract

Molecular ecology uses molecular genetic data to answer traditional ecological questions in biogeography and biodiversity, among others. Several ecological principles, such as the *niche hypothesis* and the *competitive exclusions*, are based on the fact that species compete for resources. More in generally, it is now recognized that species interactions play a crucial role in determining the coexistence and abundance of species. However, experimentally controllable platforms, which allow us to study and measure competitions among species, are rare and difficult to implement. In this work, we suggest exploiting a Molecular Dynamics coarse-grained model to study interactions among single strands of DNA, representing individuals of different species, which compete for binding to other oligomers considered as resources. In particular, the well-established knowledge of DNA–DNA interactions at the nanoscale allows us to test the hypothesis that the maximum consecutive overlap between pairs of oligomers measure the species’ competitive advantages. However, we suggest that a more complex structure also plays a role in the ability of the species to successfully bind to the target resource oligomer. We complement the simulations with experiments on populations of DNA strands which qualitatively confirm our hypotheses. These tools constitute a promising starting point for further developments concerning the study of controlled, DNA-based, artificial ecosystems.

## 1. Introduction

It is increasingly recognized that species interactions play a key role in driving the dynamics of species abundances and diversity of ecological communities [1,2,3,4,5].

In particular, a lot of effort has been put into studying how the different types of ecological interactions affect the stability of ecosystems and the ability of different species to survive in the presence of few or many resources [6,7,8,9,10].

The interactions between two species may be [11] direct (e.g., predation) or indirect (e.g., competition for a resource), may be beneficial for both species (e.g., mutualistic/cooperative interactions), or neutral for one species and positive/negative for another (e.g., commensalism/amensalism). Recent evidence also suggests that higher order (i.e., beyond pairwise) interactions may have a fundamental role in determining the stability and coexistence of diversity in ecosystems [12].

One of the challenges for testing the variety of models and predictions is the ability to measure such interactions. Competition especially, being an indirect interaction, is very difficult to measure experimentally. At the same time, strong ecological principles, such as the *niche hypothesis*, stating that the diversification of the population is a consequence of the onset of a competition for the limited available resources [13,14,15] and the related *competition exclusion*, affirming the impossibility of having a diversity larger than the number of different niches or resources, are based on competition among species. More generally, being able to quantitatively characterize the competitive advantage of a species in a given environment is crucial to defining its fitness and thus its ability to reproduce and spread in that environment. Investigating ecological competition is also challenging in a laboratory experiment, and only rarely has it been feasible to setup experimental platforms to address these questions [9,16].

A possible approach, to this extent, consists in the creation of an *ad-hoc* simulation environment in which competition mechanisms can be studied in depth. However, to do that, we need to have a paradigmatic model of *species*, where interactions can be quantitatively measured, and a clear definition of an *environment* where such species compete for one or more resources.

Each species can be characterized by its *genotype* and its *phenotype*. Following the work of Wagner [17], we define a species as individuals of a population that share a similar genotype (e.g., genotype networks) that is associated with the same phenotype [18]. Of course, such characterization of species thus depends on how we define the phenotype, that in turn depends on the ability to measure competition.

Molecular ecology [19] uses molecular genetic data or techniques to answer traditional ecological questions, such as the biodiversity characterization of a given biological sample. For examples, the existence of molecular markers conserved among species (16s rRNA gene [20]) allows the taxonomic classification of species (especially for bacteria communities [21]). Inspired by this approach, we want to exploit the known DNA–DNA interaction physics to study the ecological interactions, in particular in the competition for resources.

In this work, we suggest exploiting the OxDNA computational framework to study competitive interactions among single strands of DNA (ssDNA), which we will consider as representing individuals of different species, as detailed below. ssDNA oligomers are chains of unpaired nucleotides, highly flexible, markedly different from their much stiffer double-stranded counterpart. The great advantage of such a choice lies in the possibility of combining the knowledge of the polymer with the conceptual simplicity of the interactions among DNA filaments, aided by the large number of tools available from both the experimental [22,23,24] and computational [25,26,27] points of view. ssDNA oligomers are made by combinations of four building blocks (the nucleotides A/C/T/G) which can canonically bind in a selective way: A/T, C/G. Each ssDNA oligomer features a sequence of nucleotides (its genotype), which then fold to secondary [28] (i.e., base-pairing of complementary nucleotides) and tertiary [29] (three-dimensional configuration) structures that, as we will show, may indeed characterize the phenotype of the ssDNA.

One could then think of the simplest possible ecosystem as a paradigmatic model for more complex contexts: one species (predator) together with a single type of resource or prey (as sketched in Figure 1).

To this extent, we set two specific different ssDNA strands (each one made up by an arbitrary number of nucleotides) to be the individual belonging to the only species of the ecosystem (colored in green in the picture), and the resource (colored in red), respectively. Therefore, the phenotype of a ssDNA may be described as some measurement of the strength of its attachment to the resource, jointly with an estimation of its three-dimensional (3D) configurational structure, while binding to the resource. When a second species is added to the system (gold color), a competition between species A and B onsets in our model ecosystem. DNA oligomers thus provide an extremely simple way to test such competition mechanisms, and to highlight at the same time the connection between genotype—that is, the sequence of bases defining each element of the ecosystem—and phenotype (see the right half of Figure 1 for a visualization of the 3D structure of two simulated ssDNAs interacting and forming some Hydrogen bonds).

Here, we are able to quantitatively evaluate—from a biophysical simulation perspective—the effectiveness of bindings among those polymers. The core of our hypothesis, to be tested with simulations, is that the maximum consecutive overlap (i.e., strength of the attachment) between an individual and a resource constitutes a crucial element for competitive advantage in the ecosystem, that is, that it is a good measure of its fitness. Importantly, while it is clear that the maximum consecutive overlap gives an intensity of binding energy between oligomers, it is not trivial to state *a priori* that there is one descriptor that alone is able to provide almost complete information about the outcome of two DNA oligomers “competing” for the same resource. This work is thus conceived as a proof-of-concept of the feasibility in a wet-lab and with computer simulations of the investigation of fundamental elements of evolution in artificial systems, whose outcomes can be readily interpreted in terms of an effective metrics. To this aim, we want to investigate the secondary structure between ssDNA oligomers (i.e., identify the bases involved in optimal binding configurations) and also observe 3D configurations of such filaments, to support quantitative studies with visual analysis.

From the theoretical and computational perspective, the oxDNA coarse-grained model [25,26,27,31] provides us with a formidable magnifying glass to explore the physical processes driving to hybridization between two strands [32]. The oxDNA force field is a model of DNA at the nucleotide level, which embodies pairwise-additive forces due to the excluded volume, the phosphate backbone connectivity, the stacking, cross-stacking and coaxial stacking and hydrogen bonding between complementary base pairs [31]. Moreover, sequence-specific stacking and hydrogen bonding interaction strengths and implicit ions have been introduced in most recent versions of the model, which is capable of describing both properties of single and double stranded DNA, as well as the thermodynamics of hybridization [25,26,32]. Realistic trajectories obtained via Molecular Dynamics (MD) simulations run with the oxDNA package allow us to have both a visual snapshot of oligomers configurations and also to statistically study the binding among them, with *ad-hoc* metrics. The work is structured as follows. We first present the methodology used for our simulations, together with the overlap metrics we study in each numerical experiment. We then provide the essential details of the experimental methods we use in order to qualitatively test our theoretical prediction through molecular biology experiments. We finally present the results of our work and discuss them in light of the ecological context we have just presented. A set of conclusions and future perspectives closes the paper.

## 2. Materials and Methods

### 2.1. Simulations

The Molecular Dynamics simulations described in this work are all performed using the oxDNA software [25,26,27]; in the simulations of the larger systems described here, the GPU backend has been exploited [33]. In particular, the more refined oxDNA-2 coarse-grained model is employed in all this work, to better mimic the solvent effect and include more realistic details into the simulations [26]. The cubic simulation cell has a side of L=40 in simulation units (≈34.072 nm), in 3D Periodic Boundary Conditions. Simulations are usually run at T=313 K (by using an Anderson-like thermostat). Salt concentration is 0.24 M and the diffusion coefficient is set to the default value (2.5 in simulation units). Note that a single MD timestep (0.005 in simulation units) corresponds to ≈15 fs.

The typical simulation protocol consists of:system initialization;$5\times 10^{3}$*minimization* steps, ≈75 ps (where each strand reaches its own minimum energy configuration);*relax* phase: $10^{8}$ MD steps (≈1.5155 μs) where the system is subjected to external forces which trigger the interactions and softly enhance the formation of hydrogen bonds (HBs). In particular, strands are usually enforced to form their best attachment (see next subsection for an extended discussion concerning how we evaluate the strength of a bond between DNA strands) and are weakly attracted towards the center of the box, to prevent excessive diffusion. Temperature and volume are kept constant;*MD simulation*: $10^{8}$ MD steps where the system is not subjected to external forces anymore and freely evolves in a box at a constant volume and temperature.

The physical observables (identity of bonded base pairs, energy, particles coordinates) are only recorded for further analyses during the last stage, every $10^{3}$ steps.

The input files for simulations with the related Python scripts for analysis are provided (and regularly updated) at the GitHub repository (https://github.com/francescomambretti/stat_phys_synthetic_biodiversity accessed on 24 February 2022).

### 2.2. Overlap Metrics

To assess the strength of the interaction between two distinct ssDNAs, a metric based on the number of HBs is used in this work. As in the standard oxDNA model, we consider only canonical base–base pairings: A/T, C/G. When two strands, 1 and 2, of length *L* and *l*, respectively, are found in a relative fixed position $r_{1,2}$, it is possible to count the number of base pairings. Each strand can be represented (see Figure 2) as an array of letters, with a given orientation (either from 5${}^{\prime }$ to 3${}^{\prime }$, or vice versa), which constitutes its primary structure or genotype. With this schematization, it is possible to define a relative position $r_{1,2}$ between a strand 1 and a strand 2, for instance as the (integer) number of bases separating the two left ends of the filaments, setting the left end of strand 1 as the origin of the reference frame (in the example in the top part of Figure 2, $r_{1,2}=1$ because strand 2 is translated of a single base towards the right with respect to strand 1). The top panel of Figure 2 presents an example of two possible metrics, which are extensively used throughout the present work, based on the number of links between two ssDNAs. In fact, once two strands happen to be found in a configuration described by $r_{1,2}$, one could measure:the Maximum Consecutive Overlap (MCO${|}_{r_{1,2}}$), i.e., the largest number of consecutive matching bases (2, in the example);the Total Mixed Overlap (TMO${|}_{r_{1,2}}$), i.e., the total number of paired nucleotides in that position (3, in the example). It holds that: TMO${|}_{r_{1,2}}$≥ MCO${|}_{r_{1,2}}$, $\forall \;r_{1,2}$.

Note that, in this simplified framework, if strand 1 is made by *L* nucleotides and strand 2 by *l*, $r_{1,2}\in \left[-l+1,L-1\right]$. In Figure 2, L=11, l=8: this means that they can attach in l+L−1=18 different ways. We can thus define the optimal MCO in the whole set of MCO${|}_{r_{1,2}}$, depending on $r_{1,2}$, that is ω:(1)$$\omega :=\max_{\left\{r_{1,2}\right\}}\left(\mathrm{MCO}{|}_{r_{1,2}}\right),$$
which maximizes the MCO among all the possible MCOs corresponding to all the possible relative positions between the two ssDNAs, $\left\{r_{1,2}\right\}$ (it can be verified that the peculiar MCO of the top panel of Figure 2 coincides with ω). Note that this maximum can be degenerated, that is, it can occur in more than one position along the DNA filament. The bottom panel of Figure 2 graphically illustrates this concept, by showing what computing MCO${|}_{r_{1,2}}$ for all the $\left\{r_{1,2}\right\}$ means. This algorithm yields l+L−1 values of MCO${|}_{r_{1,2}}$, and the maximum, ω, is chosen as the best indicator of the quality of the binding between the two strands. We thus propose ω to be a key element in the interpretation of the behavior of ssDNAs, when they compete to attach to the same target, that is, to be a measure of the individual fitness.

### 2.3. Experimental Methods

#### 2.3.1. Sample Preparation

Samples were prepared with different combinations of ssDNA oligomers (1 μM each) and were diluted in 1× Saline-sodium citrate (SSC) buffer (150 mM sodium chloride, 15 mM trisodium citrate, pH 7), then were heated to 95 ${}^{\circ }$C for 5 min and allowed to cool to room temperature for 1 h before use. Possible controlling factors, affecting the duplex stability in our experiment, other than DNA sequence, were: temperature, ionic strength, and pH, all of them remaining constant throughout the duration of the experiment. The SSC buffer is one of the most common meda used in studies on DNA hybridization (Tomè et al., 2019), and the pH = 7 ensures a good DNA duplex stability (Wong et al., 2021). Regarding the thermal protocol, the 95 ${}^{\circ }$C degree incubation step allows the denaturation of possible spurious structures, while a slow cool down enables the annealing of complementary structures (Wong et al., 2021). The analyzed ssDNA oligomers have the same sequence of the ones used in the simulation analysis. The sequences (5${}^{\prime }$–3${}^{\prime }$) are: p4 (50 nucleotides), GTGCGCGTGGCAAACGGGGCGTTGTGGGGCGTGCAGCGCTGACGGTCAA; p10 (50 nucleotides), res (20 nucleotides), CGGTATTGGACCCTCGCATG. All oligonucleotides were purchased from IDT (Newark, NJ, USA). We underline that the only important requirement of our parallelism regards the target strand, which was to be shorter than the predators’ lengths, so to promote significant competition among individuals. Neither the exact number of nucleotides belonging to each sequence nor the sequences’ specific identities were fundamental, as witnessed by other measurements and simulations we performed using individuals with 50<L≤100, yielding analogous outcomes. The motivation for choosing these specific ssDNA sequences lay in a parallel experimental platform that we are currently developing and will be the subject of future publications.

#### 2.3.2. Polyacrylamide Gel Electrophoresis

A vertical polyacrylamide gel electrophoresis setup (Bio-Rad Protean II, Gel Company Inc., San Francisco, CA, USA) was used for DNA size separation. ssDNA oligomers were diluted in gel loading solution (1% glycerol), and were run on a 20% non-denaturing polyacrylamide gel (1× TAE buffer and Acrylamide/Bis-Acrylamide solution 19:1, BIO-RAD, Hercules, CA, USA), for 4 h at 90 V. The gel was stained for 15 min in 1× GelRed (Merck, Darmstadt, Germany) and was visualized on a transilluminator.

## 3. Results

We suggest that ω can be regarded as the fitness of each strand with respect to the target one, that is, that having a large ω is crucial for setting a stable and strong binding with the resource. To test this hypothesis, in both experiments and simulations, we chose a *target* strand, which mimicked the *resource* of our tiny ecosystems. The chosen resource (‘res’) was made by l=20 nucleotides and it is defined by the sequence: $$5^{\prime }\;\mathrm{CGGTATTGGACCCTCGCATG}\;3^{\prime }$$

Then, two particular sequences were selected, one designed to have a low ω with the resource (ω=4, referred to as ‘p4’), and the other one characterized by a high affinity to it (ω=10, ‘p10’).
$$\begin{array}{c} p4:5^{\prime }\;\mathrm{GGTGCGCGTGGCAAACGGGGCGTTGTGGGGCGTGCAGCGCTGACGGTCAA}\;3^{\prime }\\ p10:5^{\prime }\;\mathrm{GCGGTGCACGCAACGCCGGATGCGAGGGTGCTTGTTGCGAGGGCTGCTGG}\;3^{\prime } \end{array}$$

Importantly, as already anticipated, the only crucial condition to be satisfied is L>l, so as to enhance competition for the resource among predators; conversely, the exact choice of the bases of the sequences studied was not crucial. We tested the same ideas on ssDNA oligomers made by different bases and of different lengths, finding analogous results.

In Figure 3, the bases involved in the formation of an MCO = ω are highlighted for both predators, together with the bases contributing to the TMO, for that $r_{1,2}$. We underline that, concerning p4-res interaction, the three additional bases that originate the TMO are quite close to the four consecutive bases involved in the MCO formation. Therefore, they might play some role, maybe originating small bulges. A similar consideration can be made for the p10-res interaction, where the C-G pairing on the right is separated from the 10 bases of the MCO = ω by a single mismatch. In general, we do not expect this additional base to be highly relevant for the physics of the system. Concerning instead the p4-p10 binding, the seven additional base pairs, which are included in the TMO computation, are quite spread out along the two chains and it is inconvenient for the two filaments to form these HBs, considering the high number of mismatches interspersed among them.

These three types of oligomers were then simulated with oxDNA, in the following setups:a p4 and a resource;a p10 and a resource;two p4 strands;two p10 strands;a p4 and a p10;a p4, a p10 and a resource.

These small, controlled setups allowed us to focus on the details of two-strands (predator–prey or predator–predator) and three-strands (two predator-prey) interactions.

### 3.1. Simulation Results

For each of these cases, we ran *n* independent Molecular Dynamics simulations with the oxDNA package [25,26,27], following the protocol outlined in the Methods section. In particular, all the systems were studied over n=10 statistically independent simulations, while the system with three strands was simulated n=20 times. In each setup, the ssDNAs were equilibrated, with the aid of external forces which helped it to reach a relaxed state. In fact, during the equilibration, the two/three strands that constitute the system were usually driven to form their largest MCO (i.e., they bound in such a way that the MCO corresponding to their relative positions coincided with ω). Subsequently, the external forces were removed, and the system freely evolved at a constant volume and temperature for N = $10^{8}$ steps, undertaking an approximate, but realistic trajectory. In this way, we have a direct check: if it is energetically convenient for the strands to remain in this fine-tuned configuration, they are going to fluctuate around this equilibrium state, otherwise they are going to arrange themselves in different structures. Here follows a summary of the results concerning the strength of the inter-strand bonds, in all the simulated configurations:ω between p4 and the resource is 4. During simulations, the actual MCO is exactly equal to ω for more than 60% of the timesteps. The top left panel of Figure 4 also shows that the two strands are not bonded for roughly 25% of the time, while they can also bind and form additional HBs (with a TMO of even 8 or 9). The interaction between this predator and the target strand can be thus described as quite weak, since ω is small and even the TMO is not very large. We underline that four consecutive paired bases represent the minimum threshold for having an effective attachment between two strands: with a smaller number of bases, the two strands are not going to even bend and their HBs can be easily broken.The actual MCO between p10 and the resource is equal to ω=10 for more than 50% of the timesteps (see top right panel of Figure 4), and it is equal to ω−1 for more than 30% of the cases. Here, it is also evident that the TMO does not play a key role, compared to the MCO, since they have very similar distribution, with TMO usually being larger at most by 1 or 2. MD simulations with oxDNA suggest, thus, that the MCO onset, maybe with one additional HB formed, is the preferred way for these strands to interact. This result can be interpreted as a much stronger binding than the p4-res case.The interaction between p4 and p10, without the resource, in more than 50% of the cases occurring via the formation of an MCO =ω=7 (see Figure 4, bottom left panel). Interestingly, in about 30% of the cases, we record a TMO = 10, with a distribution of the other TMO values mainly uniformly distributed between 4 and 9 and 11 and 16 HBs. The TMO distribution features a very long right tail, indicating that rarely a huge number of nucleotides can bind between the two oligomers. In the absence of the target strand, computer simulations suggest that these two predators can bind, and this might undoubtedly alter the interaction between each of them and the resource in an environment with these three species, as we will show later in this study.If two p4 strands interact, their ω is moderately high (8), and it roughly occurs 50% of the time (see Figure 4, bottom row, middle). In about 25% of the cases, the two strands have an MCO = 4, while in about 10% they are not bonded. Interestingly, they can form very large TMOs: the TMO distribution is bell-shaped, between 6 and 19. This means that two individuals of this species may mutually attract in a significant way, possibly with a high number of HBs, in different positions along the strands.Two p10 strands have ω=6, which during the simulation occurs with a frequency slightly larger than 30%. It has to be considered that this is possibly the prevalent form of binding between these two ssDNAs, since with almost 50% probability they do not share HBs. Such an interaction between two p10 strands can be classified as weaker than the p4-p4 one.

Importantly, simulations yield analogous results even without fine-tuning the interactions during the relax phase; nonetheless, gently helping the system towards such favorable configurations is a convenient approach. Motivated by the fact that predators seemingly interact a lot with other predators of the same or other species, in order to study competition among different individuals, we prepared a setup where p4 and p10 could contend for a resource. The simulation setting and procedure is identical to the five cases previously discussed, but with three strands in the same box: one p4, one p10 and one res. p4 and p10, during the relaxed phase, were both forced to tentatively form their ω with the resource. We underline that, as shown in Figure 3, the bases ‘CCA’ of the target strand are involved in the formation of MCO =ω for both strands. Thus, if they both manage to bind to the resource, one of them is necessarily forced to form HBs elsewhere along the chain: our hypothesis is that this alternative way of binding is less favorable, yielding a competitive advantage to the predator which succeeds in setting an MCO =ω. In Figure 5, the top left panel describes the probability that p4 can be attached to the resource, in this setup with two predators and one prey. This histogram looks very different from the corresponding one of Figure 4, since in more than 80% of the simulation timesteps, p4 is not bonded at all to the resource. This is clearly due to the presence of p10 which, on one hand occupies a relevant volume fraction and, on the other hand, is more frequently bonded to the resource. Since p4 and p10 compete for the same bases (see Figure 3, ‘CCA’ bases of the resource are involved in both MCO formation), if p10 is attached to the resource, p4 is excluded and has to bind elsewhere to the resource, which happens extremely rarely. Conversely, the attachment probability between p10 and the resource is only slightly reduced by the presence of p4, which induces some repulsion effect due to its excluded volume (see Figure 4, top right panel). Here, about 25% of the time, p10 is not able to bind with res, but an MCO comprised between 7 and 10 occurs with a total probability of around 75%. A crucial element involved in the competition between predator strands is highlighted by the histogram of the interactions between them (see Figure 5, bottom panel). The ω between p4 and p10 is 7, and it is actually formed during simulations about 10% of the time, with approximately the same frequency as MCO = 5. Moreover, there are many situations when their TMO can be significantly large, thus suggesting the possible presence of interactions which bind these predators together.

### 3.2. Experimental Results with Gels

#### Polyacrylamide Gel Electrophoresis

An in-vitro approach is used to confirm the oxDNA simulation. ssDNA oligomers interactions are analyzed by polyacrylamide gel electrophoresis (Figure 6). Very interestingly, we observe gel bands corresponding to all molecular interactions predicted by oxDNA simulations. In fact, as can be seen in lane 3, the faint upper band suggests the formation of a p4 homodimer (Figure 4, bottom middle panel). In lane 4 we can detect the interaction between p10 and res oligomers. The observed smeared band can be reconducted to a different conformation of ssDNA oligomers; moreover it can also indicate the existence of alternative interactions between p10 and res. This possibility is also suggested by the oxDNA simulations where we observe multiple MCO configurations (Figure 4, top right panel). Concerning the interactions between p4 and res, no signal corresponding to a possible binding between the two oligomers is observed (Figure 6, lane 5), in agreement with the simulation (Figure 4, top left panel). A clear signal of interaction can instead be noted between p4 and p10, even though the large amount of the ssDNA oligomers does not seem to interact (Figure 6, lane 6 and Figure 4, bottom left panel). When all the ssDNA oligomers (p4, p10, and res) are mixed in the same preparation (Figure 6, lane 7), we can notice the presence of other bands in addition to the ones already described in the previous samples. In particular, two higher molecular-weight bands are present and indicate the interaction of all three ssDNA oligomers. The intense band corresponds to the p4, p10 and res complex (structure represented in Figure 7), while the upper faint band is compatible with a more sophisticated structure represented by a tetramer complex composed of two copies of res oligomer, p4 and p10 (structure represented in Figure A2).

## 4. Discussion

The simulation results exposed in the previous section, jointly with their experimental counterpart, suggest that it is indeed possible to have a first measure of the species phenotype through its maximum consecutive overlap with the resources. However, we have to consider that there are, at the experimental level, controlling factors, other than the interaction between DNA strands, affecting the duplex stability, for example, ionic strength, pH, temperature, the presence of solvents and other macromolecules within the same solution (molecular crowding) [34,35,36,37]. The latter is especially important when discussing molecular mechanisms occurring in living cells, where macromolecules occupy 20–40% of the total cell volume. In our artificial and simplified system, where only DNA molecules and salts are present in the solution, we mainly took into account the first three controlling factors, by choosing experimental conditions able to ensure optimal DNA hybridization and a good duplex stability. Nevertheless, simulation results call for further investigations, exploiting Molecular Dynamics and visual analysis tools, on other higher order structures that may contribute in determining the phenotype of our individuals.

In fact, MD simulations provide us with fundamental insights into the 3D structures of our ssDNA filaments, unveiling the presence of the hypothesized *high-order interactions*. In fact, not only can p4 and p10 bind with each other, even in the presence of the target strand, but they can also form an aggregate of three strands, together with the resource itself, as exemplified in Figure 7.

This configuration comes from a simulation with oxDNA of a p4, a p10 and a resource. At some point in the system time evolution, the p10 strand (green) manages to bind to the resource (red), and they form an MCO corresponding to their ω for most of the time. The p4 (gold), which would then—on average—lose the competition with p10 for the target (since, in order to form 4 consecutive HBs with the resource, it has to bind to some of the nucleotides already linked to p10), finds a different way to enter into the game. In fact, it binds elsewhere, at the opposite end of the p10 ssDNA filament, as a *parasite* would do. This kind of nontrivial structure not only may form spontaneously after system equilibration, but it also features a much longer lifetime (at least $6\times 10^{7}$ MD steps), with respect to a possible p4-resource attachment. Typical lifetimes of HB links (MCO (blue) and the TMO (orange)) between strands are shown in Figure 8, where the left column refers to simulations with only two oligomers, while the right one shows the same quantities for pairs of oligomers in simulations with three strands. The top row shows the MCO (blue) and the TMO (orange) formed by p4 and the resource, averaged over n=10 independent simulations, without and with the presence of p10. Noticeably, both these quantities display a neatly decreasing trend, indicating that the bond between these two strands is improbable and short-lived. In the simulations with also p10, the number of links that p4 is able to set with the resource is roughly reduced by a factor of 3 (this was already suggested by Figure 5), highlighting the competitive weakness of this predator, which is not able to bind in an enduring way to the target oligomer.

The number of paired bases between p10 and the target strand are, conversely, essentially stable for the whole duration of the simulation, fluctuating around their equilibrium values, in both cases (middle row of the picture). The presence of p4 slightly affects the number of bases that p10 is able to bind to the resource (around 7, with TMO = MCO). Nonetheless, it is apparent that p10 has a much stronger competitive advantage in binding to the target strand than p4. The bottom panel of Figure 8 displays the same quantities for the paired bases between p4 and p10, evidencing a partial equilibrium: these strands form a few HBs (approximately between 7 and 12), with a slowly decreasing trend, but the few consecutive paired bases tend to be stable in time (5 on average). The system with three oligomers experiences a reduction by a factor between 2 and 3 for both quantities (as already evident by Figure 5), suggesting that HBs between p4 and p10 may form but are not too frequent. On the other hand, once they form, they are usually maintained. This last case includes the attractive interactions between p4 and p10 both with and without the target strand, on one hand revealing the complexity of the possible interactions even in such a simplified environment, and on the other hand suggesting the possible convenience of nontrivial three body structures such as the one represented in Figure 7.

To test the validity of these results in different setups, we simulated larger systems (aided by the oxDNA GPU implementation [33]). First of all, we surrounded a p10, a p4 and a resource, whose reciprocal interactions during the relax phase were fine-tuned to be the same of the simulations previously discussed, with other 12 strands randomly displaced in the simulation cell (four for each type of predator and four resources). In this configuration, the presence of additional strands not only does not alter the already studied process (i.e., again, p10 most frequently wins the competition against p4), but also other strands reciprocally bind (and often they form an MCO very close to their ω), without any specific driving force. Even more remarkably, the stability of the binding between p10 and the resource has been confirmed by simulations with 15 strands (five for each type of predator and five resources), randomly distributed in space, without any fine-tuning during equilibration (Figure A3). Additionally, in this context, p10 predators bind more often than p4 to the resource (see Figure A4 for the pairing statistics in this case); moreover, when a p4 manages to form its ω with a resource, the life of the binding does not exceed $3\times 10^{7}$ MD steps. Conversely, p10-resource bindings last almost always at least $10^{8}$ steps. In one of such simulations, a highly complex structure involving four oligomers was detected: two resources, a p4 and a p10, all bonded together. In this peculiar case, the p10 is able to bind to one resource with MCO ω=10 and in another location to a second resource, forming a shorter MCO (eight base pairs). The other end of the p10 is exploited by a p4, interacting with it by an MCO of four bases, and forming a total of nine nucleotide pairs. Moreover, this parasite p4 is also capable of partly self-folding, with the formation of a closed loop made by those bases which do not attractively interact with p10. A visual snapshot of such a hinged configuration is reported in Figure A2 (Section A.1).

## 5. Conclusions

Here, we exploited oxDNA, a tool developed to efficiently simulate DNA–DNA interactions and DNA conformations upon hybridization, to test whether DNA–DNA interactions could constitute a simple enough platform to mimic ecological systems. In particular, we investigated competitive and cooperative interactions between DNA strands that can be considered as individuals of different species and environmental resources, the fitness being represented by the strength of their interactions. We find that, despite all the complexity of the tertiary structuring, the simple maximum consecutive overlap appears to be a sufficient quantifier of the fitness, thus opening the way to simple experimental and modeling approaches for a DNA-based model of molecular ecosystems. Interestingly, we also suggest that our results can also be leveraged for a parallelism in a biochemical context, so as to describe chemical reactions in a given initial environment, where one has reactants and catalysts competing for resources and reactions slow down as resources are depleted. However, the simulation results presented in this manuscript, strengthened by the proof-of-concept of their experimental feasibility, claim significant further extensions of the method. One may think to increase the number of species acting in such DNA-based artificial environments, adding oligomers characterized by a different genotype. Another intriguing perspective is the possibility of introducing other key elements of real evolutionary processes, such as selection (e.g., by separating individuals with a higher fitness and discarding the others) and mutation (e.g., by changing a few bases of the sequence of some ssDNA filaments). To this extent, we are currently developing an experimental platform (which leverages the well-established SELEX technique [38,39,40]) where it will be possible to mimic evolutionary cycles with a strengthening of the statistical relevance of the fittest individuals. Such experiments will be supported by computer simulations and ad-hoc Statistical Mechanics models, for a powerful description of these ecosystems populated by DNA oligomers; we believe that, also in these systems, the maximum consecutive overlap parameter will provide the best key for interpreting the results.

## Figures and Tables

**Figure 1 entropy-24-00458-f001:**
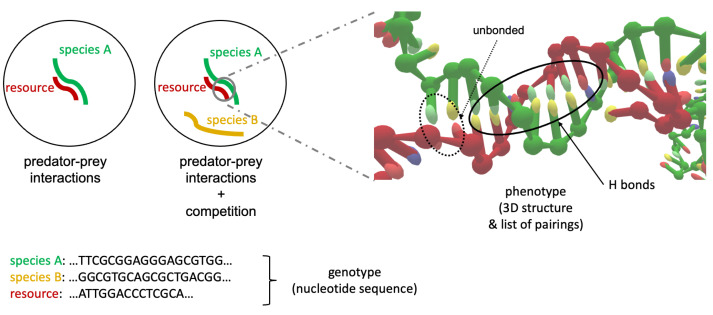
(**Left**): schematization of a paradigmatic ecosystem with a single species A interacting with a resource; aside, the presence of a second individual belonging to a different species B induces a competition between A and B for the resource. (**Right**): graphical representation (obtained with the oxView utility: https://github.com/sulcgroup/oxdna-viewer/ accessed on 24 February 2022 [30]) of the 3D configuration originated by the interaction of a predator (species A) with a resource, when they are ssDNA filaments. Bonded and unbonded base pairs are evidenced, corresponding to the secondary structure (phenotype). The sequences of nucleotides identifying each of the three strands can be thought as the genotype of each individual.

**Figure 2 entropy-24-00458-f002:**
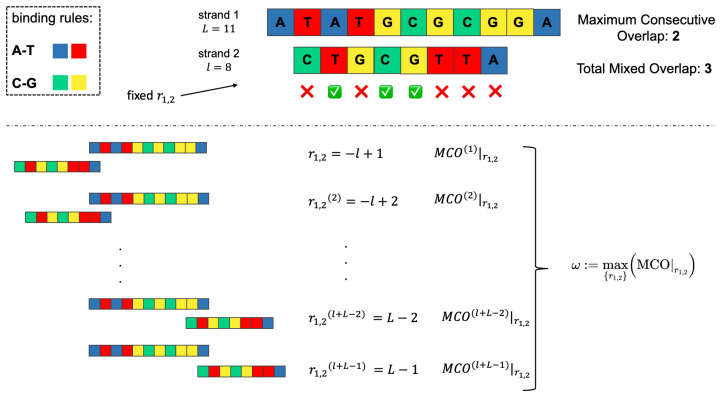
(**Top left**): binding rules for canonical DNA base pairs interactions. (**Top right**): given a strand 1 of length L=11 and a strand 2 of length l=8, schematization of the MCO and TMO between them. In this peculiar $r_{1,2}$, there is a total of three matching base pairs, but only two of them are consecutive (hence, MCO = 2 and TMO = 3). (**Bottom**): representation of the calculation of ω for strand 1 and strand 2. An MCO${|}_{r_{1,2}}$ is determined for each relative position $r_{1,2}$ between them, and then the largest is selected.

**Figure 3 entropy-24-00458-f003:**
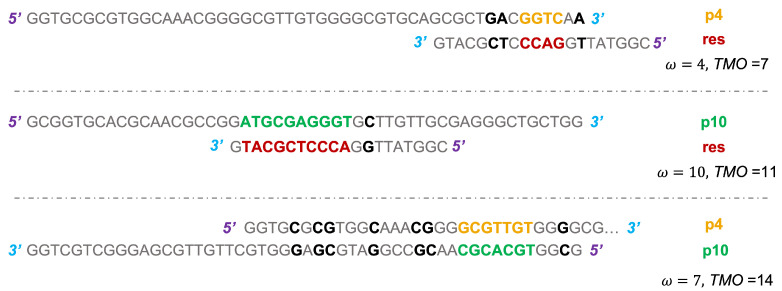
Scheme of the binding between p4 and res (**top**), where the bases forming the MCO =ω are in bold font and colored in gold (p4) and in firebrick (res). The other bases contributing to the TMO are in black color and bold font. The same applies for p10-res interaction (**middle**), where the bases of p10 involved in the formation of MCO =ω are colored in green. The (**bottom**) panel shows the configuration for the formation of MCO =ω between a p4 and a p10 strand, with the usual color scheme. The 5${}^{\prime }$ and 3${}^{\prime }$ ends are colored, respectively, in purple and cyan, highlighting the reciprocal orientation of strands.

**Figure 4 entropy-24-00458-f004:**
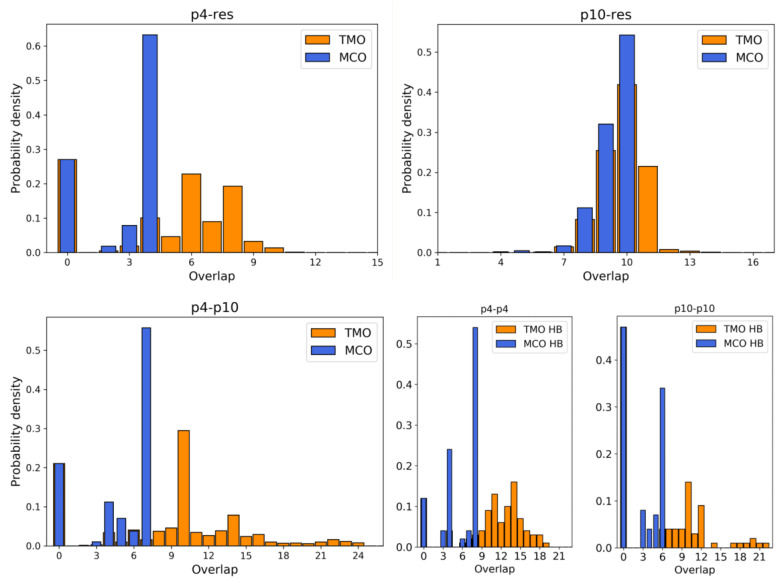
Histograms representing the probability density (obtained by cumulating the data from n=10 MD simulations) to find a given MCO (blue)/TMO (orange) between pairs of strands at equilibrium. (**Top**): p4-res (**left**), p10-res (**right**); (**bottom**): p4-p10 (**left**), p4-p4 (**middle**) and p10-p10 (**right**).

**Figure 5 entropy-24-00458-f005:**
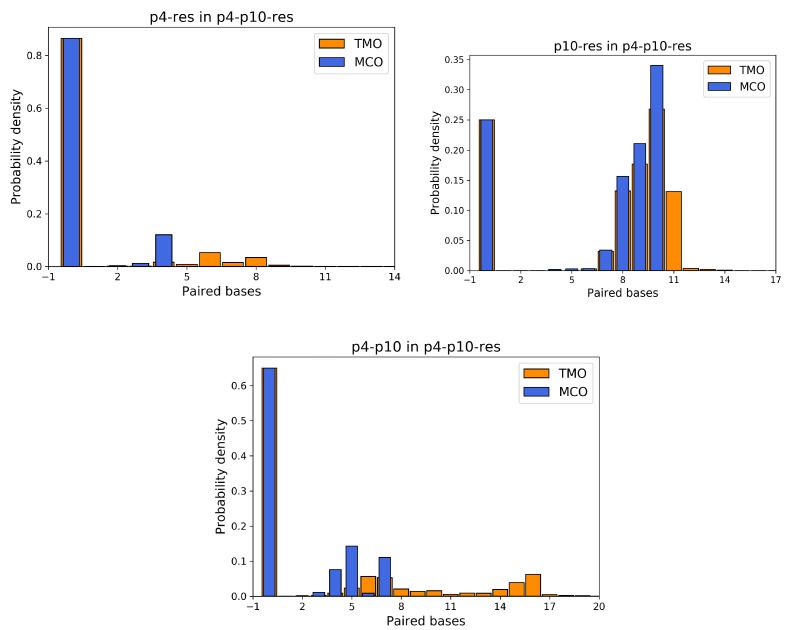
Probability densities for TMO and MCO between p4 and resource (**top left**), p10 and resource (**top right**) and p4 and p10 (**bottom**), in the simulations with three ssDNAs.

**Figure 6 entropy-24-00458-f006:**
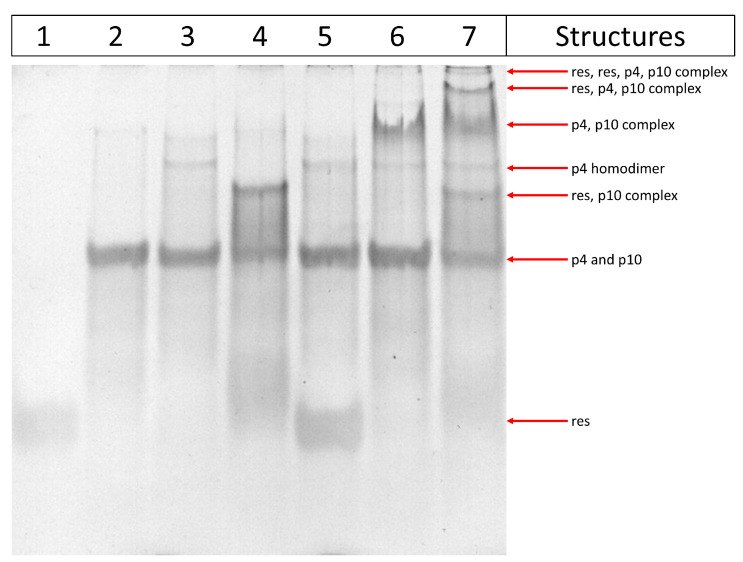
Polyacrylamide gel showing the interactions between ssDNA oligomers. Samples are prepared by mixing p4, p10 and res ssDNA oligomers, and separated on a 20% polyacrylamide gel for 4h. Each lane contains a different combination of ssDNA oligomers. Lane 1: res only; lane 2: p10 only; lane 3: p4 only; lane 4: res, p10; lane 5: res, p4; lane 6: p4, p10; lane 7: res, p4, p10. Arrows indicate gel bands and the corresponding structures formed by ssDNA oligomers.

**Figure 7 entropy-24-00458-f007:**
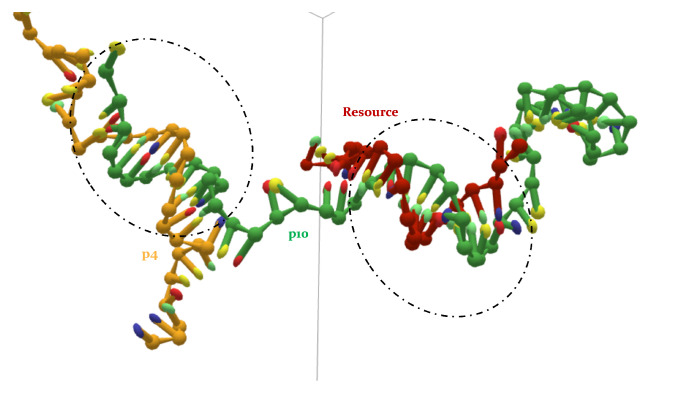
Frame from MD simulation with oxDNA, realized with oxView (https://github.com/sulcgroup/oxdna-viewer/ accessed on 24 February 2022 [30]). Example of high-order interactions: p10 (green) forms some HBs with the resource (red) and p4 (gold) binds elsewhere along the p10 strand.

**Figure 8 entropy-24-00458-f008:**
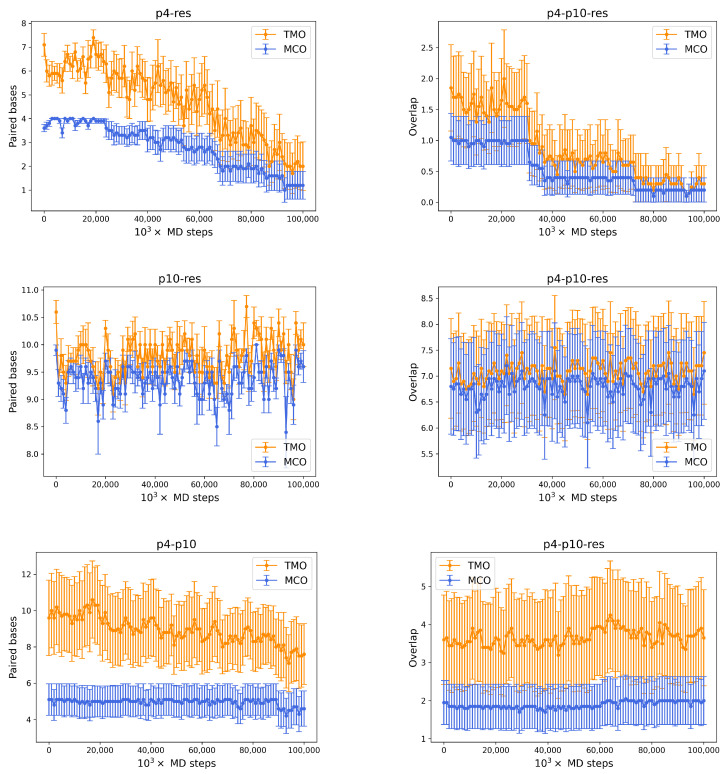
(**Left**) column: number of paired bases (TMO: orange, MCO: blue) between strands pairs as a function of time during MD simulations with p4-res (**top**), p10-res (**middle**) and p4-p10 (**bottom**). (**Right**): corresponding quantities for the simulation with one p4, one p10 and one resource.

## Data Availability

Not applicable.

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
