# Peer review of "OxDNA to Study Species Interactions"

_entropy, 2022, doi:10.3390/e24040458_

Round 1

Reviewer 1 Report

The calculations and experiments are well designed and well described. The paper is based on an unusual model (and I mean unusual in the laudatory sense) establishing/postulating a correspondence between the prey-oredator relationship and the relationship between different nucleotide sequences. What I think the authors have to add to the paper is the description of how the choice of sequences used correspond to the model of prey/predator and how this choice affects the result..

Reviewer 2 Report

The submitted manuscript presents and compares the DNA-DNA interactions at the nanoscale to investigate ecological competition between species. The subject is interesting and relevant to the Entropy. Overall, the manuscript is clear, concise, well-written, and appears to be interesting to those working in the dynamics of DNA structures. There are some important issues that should be addressed before considering this work for publication. Here are some specific remarks:

  1. The investigations are done for short DNA sequences and the choice for DNA length is not clear, i.e., p4 & p10 are of 50 nucleotides and ‘res’ is of 20 nucleotide length. The whole study depends on the sequence of these strands, I did not find any rational for these sequences of DNA strands. The significant change in sequence, whereby GC-rich region is replaced with AT-rich region, may affect these interactions. The authors should comment on how the chain length and sequence affect the formation of the duplex or triplex.

  2. Other than the DNA sequence, what are the controlling factors that decide the different formations of these complexes? should be discussed in experimental section.

  3. The authors did not discuss the role of pH in their study. It is known that the presence of salt or other biological crowders affects the base pairing between DNA strands (Physica A, 2015 Vol. 419, pp. 328-334; DOI: 10.1016/j.physa.2014.10.029 & Nucleic Acids Res. 2020 Nov 4;48(19):10726-10738; DOI: 10.1093/nar/gkaa854 & Phys. Chem. Chem. Phys. 2017, 19:19452-19460; DOI: 10.1039/C7CP03624H). It would therefore be useful to address the effect of pH or other crowders on these DNA nanostructures. I would recommend including these studies in the introduction or discussion section of the manuscript.

  4. In Figure 8, authors represent paired bases between the DNA strands. how do they make sure the base-pairing is between the strands not within the strands (formation of DNA hairpin)?

  5. In Figure A3, the authors presented a snapshot of the formation of two or more duplexes based on available p4 or p10 oligomers. However, the theoretical explanation is missing on formation of p4 & p10 duplex preference over the formation of p4- resources or p10- resources duplexes, whether it is a kinetic or entropic contribution that helps this duplex formation.

  6. What are the applications of this work as for example the authors briefly mention in the abstract, “promising starting point the study of controlled, DNA-based, artificial ecosystems.” how this process may provide more possibilities for DNA computing or a potential way to DNA-based artificial ecosystems? 

I believe that the authors should be able to address these concerns appropriately with modifications to the text. In summary, the paper may be considered for publication once the raised concerns are explained in revised version.

Reviewer 3 Report

The authors test the hypothesis that the maximum consecutive overlap between pairs of oligomers measure the species competitive advantages. Here, species, resources, environment and higher order interactions are all defined in terms of molecular interactions that can both be simulated and experimentally carried out. Nothing is surprising with respect to the physical aspects of the systems being simulated. This manuscript is well written mechanically, but for me, the purpose of this work is confusing. I have no problem following the simulations, and their experimental verification. Therefore, the results are not in question. However, the hypothesis seems out of place. If it is not true for this system, it will surely be true for other systems. The best that I can tell, the authors purposely consider a physical system at the molecular level that exhibits interesting complex self-organization, and then they map this molecular process to ecological terminology. We know that chemical reactions compete for resources (reactants and catalysts) and reactions slow down as resources get scarce. Some chemical reactions yield products that are resources for other chemical reactions. Also, the equilibrium concentrations will be dependent on environment (e.g. initial chemical concentrations, along with other thermodynamic conditions).  This complexity is present in the cellular environment, which then supports the central dogma of molecular biology, and this in turn supports macro-biological evolution, another form of self-organization.

If one accepts this paradigm, it is possible to make analogies, and moreover even understand the root cause of all this self-organization: the production of entropy.  The authors state their goal (or at least what I understood their goal to be) is to simulate macro-ecological evolution in a molecular level system. I think the analogies are strong enough that the outcome of this work can be expected, and the almost contrived system had to work. Hence, the hypothesis could not be proved wrong, even if it took a different system (for resource molecules to bind to species molecules). Hence, this is not a valid hypothesis because it cannot be wrong. In my view, the authors are trying to show a 1-to-1 mapping between physical interactions at a molecular level, where we have strong theory to understand self-organization, to the macro-ecological evolutionary dynamics, which is impossible to experimentally track on timescales of a human life span.

I do not see this work producing anything more than a nice analogy. If the goal is to find a way to simulate large scale systems to model macro-ecological evolutionary dynamics, this is best done with agent-based modeling. On the other hand, if the goal is to develop experimental methods to do molecular-ecological experiments, to test long time results of ecological evolution that could take billions of years to occur, but in a wet-lab, only days or weeks, this is interesting. However, I think the authors are not making this clear.

I do not have a strong opinion about whether the manuscript is ready for publication as it is technically correct. However, as a reader who appreciates connections between different areas of research, I am not getting anything out of the conclusion or believe the hypothesis is special.  To increase enthusiasm for these results, after the authors read my take on the motivation for their work, perhaps they can improve the abstract, introduction and conclusions to make their goals clearer so others can share their enthusiasm.

Round 2

Reviewer 2 Report

The authors appropriately took all criticism into account. The manuscript is more reach in context of understanding the DNA-DNA interactions at the base pair level. This study would be of interest for the scientific community. My recommendation is to accept this work.

Reviewer 3 Report

The authors response to my concerns made more clear the motivations of the work, and showed the hypothesis is falsifiable and the authors made minor edits to reflect these concerns in a sufficient way, per my suggestions.